# Readiness of health posts to manage child wasting in six districts of Ethiopia

**Alinoor Mohamed Farah** [1,2], **Samson Gebremedhin**[1], **Beshada Rago**[1], **Aweke Kebede**[3], **Kemeria Barsenga**[3], **Tafara Ndumiyana**[3], **Tayech Yimer**[3], **Hiwot Darsene**[4], **Shibru Kelbessa**[4], **Beza Yilma**[1], **Seifu Hagos Gebreyesus** [1]*

1 School of Public Health, College of Health Sciences, Addis Ababa University, Addis Ababa, Ethiopia, 2 Department of Public Health Nutrition, School of Public Health, College of Medicine and Health Sciences, Jigjiga University, Jigjiga, Ethiopia, 3 World Food Programme (WFP), Addis Ababa, Ethiopia, 4 Nutrition Coordination Office, Federal Ministry of Health, Ethiopia

* seif_h23@yahoo.com

## Abstract

### Background

Service readiness is essential for providing effective health services. Adequate infrastructure, trained staff, and essential commodities must be in place to enable timely diagnosis and treatment, without which improvements in child survival cannot be achieved.

### Objective

To assess the readiness of health posts to manage child wasting in six selected districts.

### Data and methods

A facility-based cross-sectional survey was conducted in April 2023 to assess the readiness of 72 randomly selected health posts in six districts. A composite index was developed using a weighted additive approach to assess the availability and readiness of severe acute malnutrition (SAM) and moderate acute malnutrition (MAM) services. Readiness scores were defined as the weighted mean of the availability of basic amenities, staff and guidelines, equipment and medicine, and nutritional commodities. Descriptive statistics were used to report the availability of tracer items, and the second (median) and third quantile regression models were used to analyze the association between independent variables.

### Results

The overall SAM and MAM readiness scores were 57.9% and 53.9%, respectively. Basic amenities scored the lowest (45.8% for both) owing to limited electricity

**Data availability statement:** Data for this study cannot be shared publicly because it contains sensitive information which may allow the identification of specific individuals and facilities, particularly the health workers and health facility geolocations, who participated in this research. Data are available upon request from the College of Health Sciences, Addis Ababa University, Research Ethical Review Board via email (chs.irb@aau.edu.et) for researchers who meet the criteria for access to confidential data.

**Funding:** The current study is supported by World Food Program (WFP), Addis Ababa, Ethiopia.

**Competing interests:** he authors have declared that no competing interests exist.

**Abbreviations:** CMAM: Community Management of Acute Malnutrition; IMAM: Integrated Management of Acute Malnutrition; EDRMC: Ethiopia Disaster Risk Management Commission; FMOH: Federal Ministry of Health; GAM: Global Acute Malnutrition; MAM: Moderate Acute Malnutrition; SAM: Severe Acute Malnutrition; WFP: World Food Program; WHO: World Health Program.

(40.3%) and water access (51.4%). Staff and guideline readiness was 66.7%, with highly trained staff (88.9%) and registration/follow-up cards (90.3% SAM; 86.1% MAM), but low growth charts (11.1%) and IYCF aids (47.2%). The availability of equipment was 60.3% for SAM and 64.6% for MAM, with high MUAC tapes (97.2%) and scales (83.3%), but low timers (16.7%). The availability of medicines and commodities scored 67.1% for SAM and 69.0% for MAM, with RUTF/RUSF (79.2% SAM; 59.7% MAM) and vitamin A (77.8%); gaps remained for antibiotics (55.6%), zinc (61.1%), and IFA tablets (58.3%). Only 19.4% of facilities met the ≥75% readiness for SAM, compared with 11.1% for MAM.

## Conclusion

In conclusion, improving wasting service readiness requires strengthening basic infrastructure, ensuring reliable supply chains, and investing in training and supervision to build MAM capacity. The wide variation in facility preparedness highlights the need for context-specific approaches, with tailored support for remote and crisis-affected areas to achieve effective integration of MAM services into primary health care.

## Introduction

Acute malnutrition, also known as wasting, occurs when a child's nutritional status rapidly deteriorates, causing them to become too thin for their height due to poor nutrient intake and/or disease. Weight-for-height, mid-upper arm circumference (MUAC), and nutritional edema are used to define wasting [1]. The two forms of wasting are severe acute malnutrition (SAM) and moderate acute malnutrition (MAM). Global acute malnutrition (GAM) refers to the combination of MAM and SAM and is often used to measure nutritional status at the population level, indicating the severity of an emergency [2].

The introduction of the Community Therapeutic (CTC) and the subsequent Community-based Management of Acute Malnutrition (CMAM) in the late nineties had a significant impact on the humanitarian sector [3,4]. These innovations were designed to be delivered in emergency contexts, and UNICEF was responsible for providing outpatient care for uncomplicated SAM, which included the distribution of ready-to-use therapeutic food (RUTF). The World Food Programme (WFP) was responsible for implementing targeted supplementary feeding programs (TSFPs) for MAM [5].

As the primary United Nations health regulatory organization, the World Health Organization (WHO) has historically been responsible for addressing the needs of children with SAM. As the WHO provides guidance specifically for the treatment of SAM, the focus is primarily on complicated SAM cases. Prioritizing children at the greatest risk of mortality, the CMAM approach initially focused on treating SAM, as children with SAM have a 9–12 times higher risk of death than well-nourished children [6], as evidenced by the 2007 joint United Nations statement on

community-based SAM management [7]. Since then, there has been a significant push, mainly led by UNICEF, to implement community-based SAM treatment within existing healthcare systems. Consequently, the treatment model for outpatient SAM management has continued to evolve. Unfortunately, this progress has not been matched by an equal focus on scaling up the supplementation and prevention of MAM [8]. Thus, the current study assessed the readiness of health posts to provide services for children with SAM and MAM.

Ethiopia was one of the first countries to pilot the CMAM model in the early 2000s. The country has since embarked on a massive roll-out of the CMAM Program in the HEP [9], which is now delivered in over 18,000 facilities [10]. HEP has enabled Ethiopia to achieve significant improvements in maternal and child health and prevent child death through timely admission and treatment of wasting [11].

Since 2020, the CMAM community has focused on simplifying CMAM to expand coverage by testing MUAC-based and combined protocols, such as ComPAS and OptiMA, which have shown non-inferior recovery compared with standard protocols [12,13]. Operational studies have reported promising gains from these simplified models but have also highlighted persistent challenges, including relapse and retention, limited reach in remote and pastoral areas, and supply chain constraints [14,15]. To address these operational gaps and validate effectiveness at scale, ongoing trials in Ethiopia are testing optimized dosages and delivery approaches [16].

The extent to which a country's healthcare system can handle the management of wasting is a crucial factor in maintaining continuity of care [8]. Over the last 13 years (2008–2020), UNICEF has made significant progress in strengthening health systems and integrating SAM treatment into Ethiopia's primary healthcare, resulting in an estimated 3.6 million SAM children being enrolled in the CMAM program [10]. However, there were no similar goals for the management of MAM until August 2019, when the Ministry of Health (MoH) released a new National Guideline for the Management of Acute Malnutrition, which integrated MAM management into the Ethiopian healthcare system [17]. Prior to 2019, the National Disaster Risk Management Commission (NDRMC) oversaw the management of MAM and TSFP with the support of the WFP, providing monthly food rations based on screenings conducted by Health Extension Workers (HEWs) [18].

Integrating the Management of MAM into primary healthcare has the potential to increase the caseload, which could compromise the care provided to SAM children and thereby further stretch already limited resources. Therefore, it is crucial to determine to what extent health posts in pilot IMAM districts prepared to manage SAM and MAM. Therefore, knowing the readiness of health posts to provide both SAM and MAM services will support improvements in services and guide the design and implementation of strategies to enhance the quality of care provided.

## Methods

### Study setting

Within Ethiopia's primary healthcare framework, services are delivered through a tiered structure of primary hospitals, health centers, and five satellite health posts per catchment, each staffed by two HEWs serving roughly 4,000–5,000 rural residents. Launched in 2003, the HEP set out to achieve universal primary-care coverage for Ethiopia's rural population by equipping HEWs to provide essential promotive, preventive, and limited curative services via both outreach and facility-based activities. Management of acute malnutrition both SAM and MAM has been adopted as a core responsibility within this HEP platform [19].

Integrated Management of Acute Malnutrition (IMAM) is currently being piloted in 150 districts across six regions: Afar, Amhara, Oromia, Sidama, South Ethiopia, and Somali. For this study, health posts offering IMAM services in six districts, namely Mile, Delanta, Grawa, Aleta Chuko, Zala, and Adadle, from the regions were selected in consultation with relevant stakeholders, including the MoH. Delanta district is in the Amhara Region, 499 km from Addis Ababa. Grawa district is in Oromia Regional State's Eastern Hararghe Zone, 529 km east of Addis Ababa. Aleta Chuko district is in Sidama region, about 275 km from Addis Ababa. Zala district is in SNNPR, 214 km south of Addis Ababa. Adadle is one of the districts in Somali Region. The prevalence of wasting varies across regions in Ethiopia, with the highest rates observed in Afar

(26%), Somali (17%), and Amhara (15%). In contrast, Oromia, Sidama, and South Ethiopia report lower prevalence rates of less than 15% [20].

In IMAM districts, MAM management is integrated into routine health services. MAM treatment is coordinated with SAM treatment, particularly in OTP, to ensure a continuum of care for acute malnutrition. Health extension workers at health posts conduct screenings, admissions, and follow-ups. HEWs perform follow-ups every two weeks for a maximum of four months, during which medical history and anthropometric measurements are taken to monitor progress.

### Study design and data collection period

A facility-based cross-sectional survey was conducted in April 2023. The readiness of health posts and provision of acute malnutrition services were assessed across 72 health posts in six selected districts (12 health posts per district).

### Sample size and sampling procedure

The study was conducted in six IMAM districts. The six IMAM districts were selected in consultation with the MoH, WFP, and UNICEF. The main selection criterion was their classification as priority 1 hotspot districts for food insecurity at the time of the survey. The structural availability and readiness assessment (SARA) was conducted in 72 health posts. Each district had four PHCUs, and three health posts were randomly selected from each PHCU using the lottery method, resulting in a total of 12 health posts per district. The study participants were HEWs at health posts, with information gathered mainly through observation.

### Data collection

The Service Availability and Readiness Assessment (SARA) tool was adapted for this purpose [21], with questions covering facility infrastructure, availability of staff and guidelines, essential equipment, and the provision of medicines and nutritional commodities. The data collection approach involved reporting the presence of specific structures, commodities, and interviewers and confirming their availability and functionality on the survey day. The indicators used to create the indices will be binary. Each indicator will be coded as 1 if present (for structures/readiness) and 0 if absent.

The checklist was adapted to be consistent with the Ethiopia wasting management protocol. The equipment included weighing scales, tape measures for measuring mid-upper arm circumferences (MUAC tapes). The guidelines included a quick translated reference guide for OTP and TSFP, look-up tables/charts, registration book for SAM, registration book for MAM, Outpatient Therapeutic Program (OTP) treatment and follow-up card/MAM follow-up card, TSFP identification card, and infant and young child feeding counselling guidelines/job-aids/quick reference. To ensure content and face validity, the adapted tool was reviewed by technical experts from the MoH and Addis Ababa University.

Before conducting any observations, informed verbal consent was obtained from the health worker. Data collectors explained the purpose of the observation, reassured participants that the emphasized that no personally identifiable information would be recorded. The data were trained to be as unobtrusive as possible to minimize disruption to routine care.

Survey data was conducted by 12 trained and experienced enumerators and six supervisors. Recruitment of these data collectors was based on multiple criteria, including educational status (at least diploma holders in health-related disciplines), experience in similar surveys, previous experience of collecting data using ODK, and successful completion of the data collectors' training. Prior to deployment, the enumerators and supervisors received four days training.

The tool was translated into local languages (Amharic, Afan Oromo, and Somali). The data were collected using the Open Data Kit (ODK), an open-source user-friendly application system. The ODK platform ensures close to real-time quality data collection, cleaning, and monitoring. The data were uploaded daily by the enumerators to the ODK cloud server.

## Index creation

A composite index is developed using a weighted additive approach. The weighted additive index was created by summing the elements and considering the indicator counts within each domain. This process included adding the indicators in each domain, dividing the sum within each domain by the number of indicators in that specific domain (Table 1), multiplying by 100, and finally dividing by the total number of domains within the index. Previous research has demonstrated that employing three different techniques, simple additive, weighted additive, or principal component analysis, to compute quality indices typically yields results that are generally consistent [22]. The weighted additive method is recommended owing to its simplicity of calculation and interpretation [23].

$$Y_{weighted\ addictive} = \frac{\sum_{d=1}^{m} \left( \frac{\sum_{j=1}^{m} x_{dj}}{n_d} \right) \times 100}{m}$$

## Data analysis

The data collected allowed us to assess the availability and readiness of SAM and MAM services for children under five. Readiness scores were defined as the weighted mean of the availability of basic amenities, staff and guidelines, equipment and medicine, and nutritional commodities. For the present analysis, the readiness score was categorized as low (<75%), intermediate (75–99%), and ready (100%), based on the weighted score capturing the availability of basic amenities, staff and guidelines, equipment and medicine, and nutritional commodities [24].

## Statistical analysis

First, descriptive statistics were used to report the availability of the tracer items using absolute frequencies. The readiness percentage score was computed based on the availability score of tracer items for each service, namely, SAM and MAM. Additionally, the second (median) and third quantile regression models were applied to analyze the association between independent variables (agrarian/pastoral areas and proximity to catchment health centers). The Afar and Somali regions were classified as pastoral, as much of the population depends on livestock and nomadic livelihoods, while the remaining regions were classified as agrarian, where settled farming is the predominant livelihood. The distance to the catchment health center was categorized as <20 or ≥20 km. Simple and multiple quantile models were chosen because the assumptions used to perform linear regression models were not satisfied by our data [25]. The Wilcoxon signed-rank test was used to compare the distributions of readiness by service type stratified by area and distance. Readiness was summarized as median and interquartile intervals (IQ), that is, Q1–Q3. STATA version 17 was used to conduct descriptive statistics and inferential analyses. P-values of < 5% were considered statistically significant.

**Table 1. Summary of domains and indicators used to create indices for readiness.**

| Index | Domain | Number of indicators/tracer items | | Source of data |
|---|---|---|---|---|
| | | SAM | MAM | |
| Service readiness | Facility infrastructure | 2 | 2 | Facility |
| | Staff guidelines | 7 | 8 | Facility |
| | Equipment | 5 | 4 | Facility |
| | Medicine and commodities | 7 | 3 | Facility |

### Operational definitions

**Readiness.** Readiness was defined as the availability of the minimum required amenities, trained staff and guidelines, essential equipment, and medicines/commodities needed to deliver SAM and MAM services. A facility was considered ready if it achieved a score of ≥75% across these domains.

**Tracer items.** Are specific, observable indicators used to assess service readiness. They are chosen because their presence (or absence) reflects whether a facility can effectively deliver a service. For SAM and MAM readiness, tracer items fall into five domains (amenities, trained staff and guidelines, essential equipment, and medicines/commodities).

### Ethical consideration

The health facilities survey was designed and carried out with the utmost respect for the rights of study participants, upholding principles of research ethics such as justice, beneficence, and non-maleficence. This study was conducted in compliance with both national and international ethical guidelines, with approval from the Institutional Review Board (IRB) of the College of Health Sciences, AAU. Necessary administrative clearances were obtained from MoH and relevant regional health bureaus. Data were collected after obtaining informed consent from the health care providers.

## Result

### Availability and readiness

The readiness assessment showed varying levels of preparedness across service domains for SAM and MAM management (Table 2). For basic amenities, the weighted score was 45.8% for both SAM and MAM, driven largely by the low availability of electricity (40.3%) and limited access to improved water sources (51.4%).

In the staff and guidelines domain, SAM and MAM scored 66.7%. The high coverage of trained staff in the management of acute malnutrition (88.9%) and the availability of registration and follow-up cards (90.3% for SAM; 86.1% for MAM) supported this moderate readiness. However, gaps in growth charts (11.1%) and IYCF counselling aids (47.2%) were observed.

For equipment, the weighted scores were 60.3% for SAM and 64.6% for MAM, with widespread availability of MUAC tapes (97.2%) and child/infant scales (83.3%), however, low availability of timers (16.7%) were observed.

The medicines and commodities domain scored 67.1% for SAM and 69.0% for MAM, supported by the good availability of RUTF/RUSF (79.2% for SAM; 59.7% for MAM), vitamin A capsules (77.8%), and deworming tablets (69.4%). Low availability of antibiotics, zinc, and IFA tablets in SAM services were also observed.

Overall, the average weighted readiness score was 57.9% for SAM and 53.9% for MAM, indicating that SAM services were slightly better prepared. Only 19.4% of facilities met the ≥75% readiness threshold for SAM, compared with 11.1% for MAM. Fig 1 illustrates the overall weighted readiness scores for SAM and MAM services across the health facilities. In general, SAM services had a slightly higher median readiness score (55.6%, IQR: 45.8 to 70.5) compared with MAM services (53.2%, IQR: 41.9 to 67.1). A statistically significant difference was observed between SAM and MAM service readiness across facilities ($p < 0.001$), indicating that readiness for SAM services was higher, although both services exhibited substantial gaps. Table 3 provides detailed descriptive statistics.

Fig 2 illustrates the differences in service readiness scores between the agrarian and pastoral areas. In pastoral regions, both SAM and MAM services had low levels of readiness, with the median readiness score highest for SAM (median (IQ): 50.9 (40.8 to 80.9)) and MAM (median (IQ): 40.0 (28.6 to 49.5)), with a statistically significant difference between SAM and MAM ($p < 0.001$). in other words, SAM management service is better prepared than MAM services. In agrarian regions, service readiness was higher for both SAM and MAM, with median scores of 61.6 (49.1 to 71.7) for SAM and 60.5 (47.5 to 67.3) for MAM, and statistically significant difference between the readiness levels of the two services ($p = 0.047$).

**Table 2. Availability of items and readiness for SAM and MAM management.**

| Tracer items | | SAM % (n/N) | Weighted Readiness score (Unweighted) | MAM % (n/N) | Weighted Readiness score (Unweighted) |
|---|---|---|---|---|---|
| Basic amenities | Electricity | 40.3(29/72) | 11.5(45.8) | | 11.5(45.8) |
| | Improved water source | 51.4(37/72) | | | |
| Staff and guidelines | Quick translated reference guide for OTP and TSFP | 73.6(53/72) | 16.7(66.7) | 73.6(53/72) | 14.3(66.7) |
| | Staff trained in diagnosis and management of acute malnutrition (SAM and MAM) | 88.9(64/72) | | 88.9(64/72) | |
| | Look up tables/charts available | 59.7(43/72) | | 59.7(43/72) | |
| | Registration book for SAM treatment/Registration book for MAM | 90.3(65/72) | | 86.1(62/72) | |
| | Outpatient Therapeutic Programme (OTP) Treatment and Follow-up Card/ MAM Follow-up Card | 95.8(69/72) | | 88.9(64/72) | |
| | TSFP Identification Card | – | | 77.8(56/72) | |
| | Growth charts (WHO) | 11.1(8/72) | | 11.1(8/72) | |
| | Infant and Young Child Feeding counselling guideline/job-aids/quick reference. | 47.2(34/72) | | 47.2(34/72) | |
| Equipment | Child and infant scale | 83.3(60/72) | 13.0(60.3) | 83.3(60/72) | 12.3(64.6) |
| | Weight scale | 37.5(27/72) | | 37.5(27/72) | |
| | Thermometer | 66.7(48/72) | | – | |
| | MUAC tapes for U5 children | 97.2(70/72) | | 97.2(70/72) | |
| | Timer | 16.7(12/72) | | 16.7(12/72) | |
| Medicine and commodities | RUTF/RUSF | 79.2(57/72) | 16.8(67.1) | 59.7(43/72) | 15.9(69.0) |
| | Oral rehydration solution packet | 68.1(49/72) | | – | |
| | Amoxicillin (dispersible tablet 250 or 500 mg OR syrup/suspension) | 55.6(40/72) | | – | |
| | Vitamin A capsules | 77.8(56/72) | | 77.8(56/72) | |
| | Me-/albendazole cap/tab | 69.4(50/72) | | 69.4(50/72) | |
| | Zinc sulphate tablets, dispersible tablets, or syrup | 61.1(44/72) | | – | |
| | IFA tablets | 58.3(42/72) | | – | |
| | **Overall weighted readiness score (mean, CI 95%)** | | **57.9 (54.1, 61.7)** | | **53.9 (49.7, 58.1)** |
| | **Facilities with weighted readiness score ≥ 75%** | **19.4(14/72)** | | **11.1(8/72)** | |

Fig 3 presents the differences in the service readiness score according to the health post distance from catchment health centers. Among health posts close to catchment health centers (<20 km), the readiness percentage score was highest for SAM median (IQ): 63.4 (50.3 to 73.5)) at $p = 0.043$. Both SAM and MAM services have low levels of readiness. Among health posts far from the catchment health posts (>20 km), the service readiness percentage score was highest for SAM (median (IQ): 51.8 (41.1 to 59.2)) at $p < 0.001$.

## Quantile regression model estimates for readiness

Table 4 presents the median and third quantile regression model estimates for readiness for SAM and MAM services. For SAM readiness, neither the area (agrarian and pastoral) nor proximity to the catchment health center significantly influenced readiness. For MAM readiness, agrarian areas were positively associated with higher readiness for MAM ($p = 0.040$), and readiness was more pronounced in the context of higher levels of readiness ($p = 0.02$). Furthermore, being

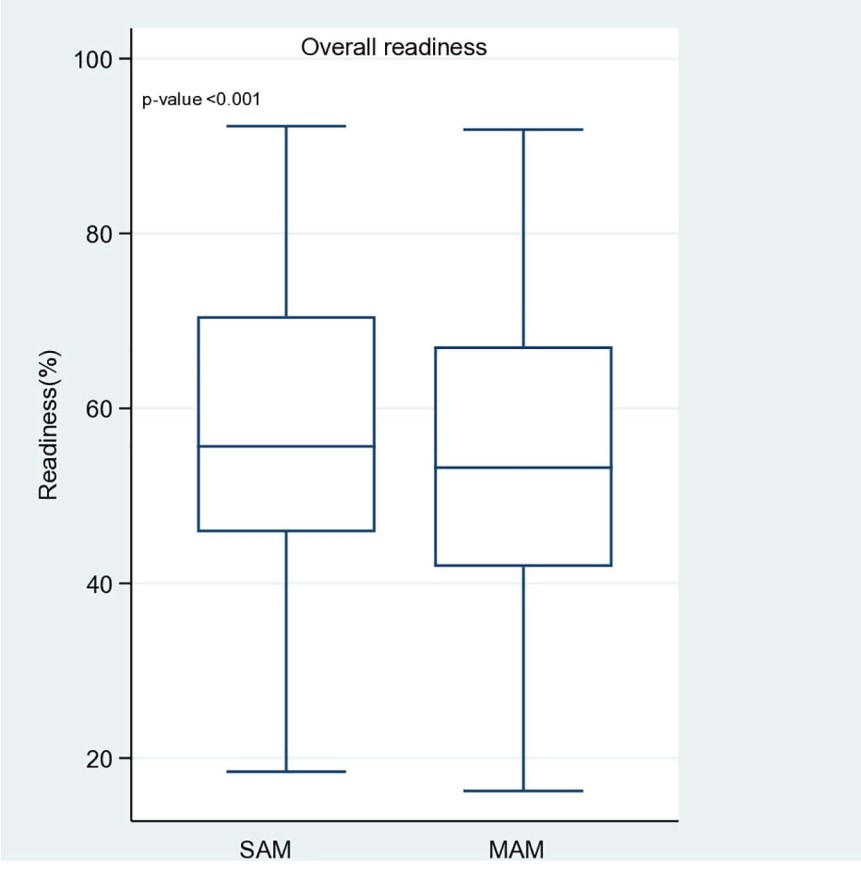

**Fig 1. Readiness scores for SAM and MAM services.**

closer to the catchment health center was also associated with higher readiness ($p = 0.016$). However, the effect of proximity is not significant in the context of higher readiness.

## Discussion

The readiness assessment showed moderate overall preparedness for wasting services in the health posts surveyed, with notable gaps across domains. Basic amenities scored poorly (45.8% for both SAM and MAM), whereas staff/guidelines and medicines/commodities performed better (66.7% and 67.1% for SAM, 66.7% and 69.0% for MAM, respectively). SAM services were slightly more prepared than MAM services, with overall weighted scores of 57.9% versus 53.9%, and only 19.4% of facilities achieved ≥75% readiness for SAM compared with 11.1% for MAM. These findings align with the evidence from other low- and middle-income countries that documented infrastructure deficits, irregular supply chains, and uneven workforce capacity as dominant constraints on facility-level malnutrition response [24,26–29].

For example, in Bangladesh, facilities lacked essential anthropometric tools, glucometer, medicines, and therapeutic foods, with inadequate manpower and training being major barriers [28]. Similarly, an assessment in India found deficiencies in helper staff, medical social workers, and emergency equipment, though beneficiaries expressed satisfaction with available services [26]. Mozambique's health facilities showed poor readiness scores, with nutrition services scoring 57.1% and diarrhea services 72.2%, primarily due to unavailable human resources, guidelines, and training [24].

**Table 3. Descriptive statistics of readiness according to the type of area and distance from the catchment health.**

| Readiness type | N | Min | Q1 | Median | Q3 | Max |
|---|---|---|---|---|---|---|
| Overall readiness | | | | | | |
| SAM | 72 | 18.4 | 45.8 | 55.6 | 70.5 | 92.3 |
| MAM | 72 | 16.3 | 41.9 | 53.2 | 67.1 | 91.9 |
| Readiness in pastoral regions | | | | | | |
| SAM | 24 | 18.4 | 40.8 | 50.9 | 80.9 | 92.3 |
| MAM | 24 | 16.3 | 28.6 | 40.0 | 49.5 | 91.9 |
| Readiness in Agrarian regions | | | | | | |
| SAM | 48 | 33.3 | 49.1 | 61.6 | 71.7 | 88.1 |
| MAM | 48 | 27.7 | 47.5 | 60.5 | 67.3 | 90.0 |
| Distance from catchment health center (<20 km) | | | | | | |
| SAM | 40 | 35.1 | 50.3 | 63.4 | 73.5 | 92.3 |
| MAM | 40 | 32.5 | 48.8 | 59.9 | 67.7 | 91.9 |
| Distance from catchment health center (>20 km) | | | | | | |
| SAM | 32 | 18.4 | 41.1 | 51.8 | 59.2 | 84.5 |
| MAM | 32 | 16.3 | 30.2 | 45.3 | 62.6 | 86.9 |

The notably low score for basic amenities (electricity and water) mirrors multi-country analyses that identified infrastructure as a recurrent weak link in delivering quality care [27]. This may also be explained by the fact that our study was conducted at lower levels of primary healthcare, health posts in rural areas where facilities often lack essential resources and infrastructure. Similar findings have been reported elsewhere; for example, one study showed that hospitals generally had much better service readiness compared with health centers and clinics [30]. Primary healthcare is a vital source of essential services for underserved populations and has the potential to serve as a powerful platform for addressing a wide range of health challenges in low-income countries [31]. However, this potential can only be realized if greater attention is directed toward strengthening infrastructure and supplies at health posts, which are the first point of care for much of the population, especially in rural areas.

Providing child wasting services requires a reliable supply chain that ensures the availability of essential nutritional products and medicines needed for service delivery. It is crucial to manage SAM and MAM by providing RUTF or RUSF, which depend on the supply chain's ability to transport these products from central warehouses to individual health facilities. Stockouts can hinder access to care and discourage community members from seeking these services [32]. It is also worth mentioning that some challenges are unique to specialized nutrition products designed to manage wasting in addition to health system limitations, including limited human capacity, infrastructure, and transportation to hard-to-reach regions. It is crucial to consider specific challenges, including the considerable size and volume of the commodity, the unpredictability of demand due to seasonal and emergency-related fluctuations, and the potential for product misuse [33].

Despite the considerable training efforts made by the nutrition partners, the study results indicated that a substantial portion of the health workforce in the study areas had not been trained in the new guideline for managing acute malnutrition, with 12% of the workforce remaining untrained. This lack of training may impair healthcare facilities' readiness to provide wasting services. Frequently, pre-service and in-service health training curricula lack a nutrition component that includes the significance of wasting management, how to detect wasting, where to refer, and other relevant information. The combination of inadequate training and the limited time that healthcare workers have with each beneficiary contributes to the under prioritization of wasting services [34–36].

Although both SAM and MAM readiness scores were low, we observed a significant difference between the two services, with SAM outperforming the MAM. One possible explanation for this difference is that SAM services have already

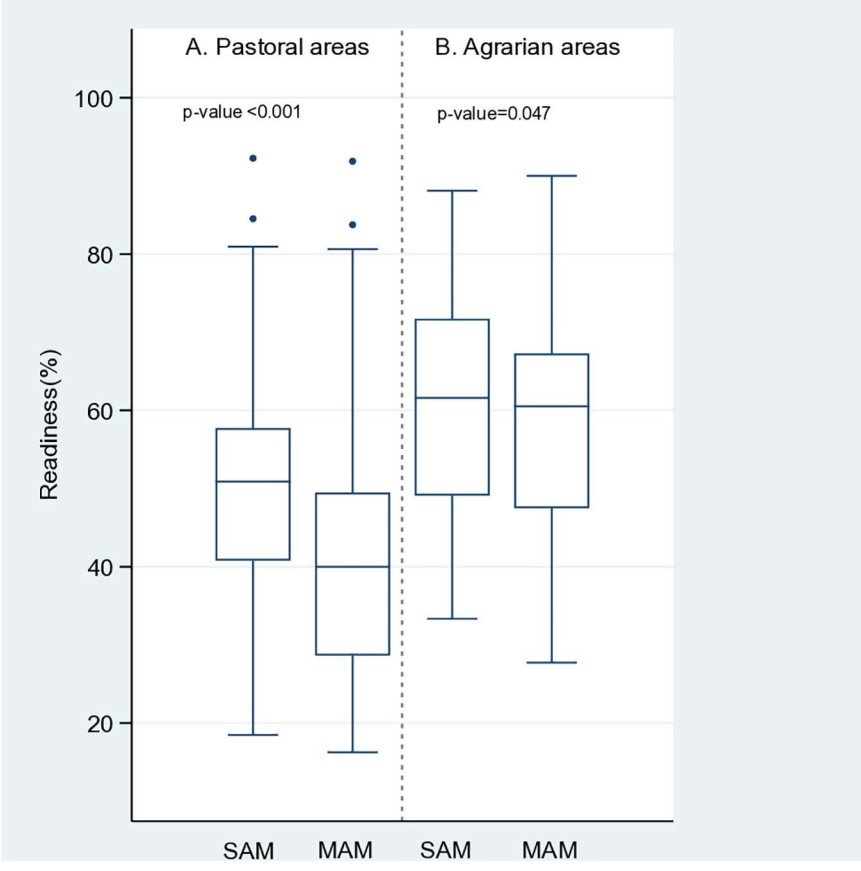

**Fig 2. Service readiness according to the type of area.**

been integrated into the health system (since 2008), while MAM is still in its early stages of integration into Ethiopian primary health care. While global guidelines for SAM exist and have provided relatively consistent national guidelines, leading to successful integration into health systems [2,37,38], MAM lacks equivalent models for scaling up, including health system integration. As a result, a vacuum has been created, which has been filled by varied national and agency policy guidance and practices, often biased towards humanitarian settings [8,39].

The institutional separation of responsibilities between UNICEF and WFP poses challenges stemming from their respective ways of working. For instance, WFP TSFPs are designed to address GAM levels that exceed 10% or are between 5–9% with exacerbating factors that primarily relate to food security [2,40]. By contrast, UNICEF aims to achieve 100% coverage for SAM treatment, particularly in areas where GAM rates are high and health services are available. It is worth noting that SAM services are provided throughout the year [41], whereas TSFPs may only be delivered during vulnerable periods, which typically span part of the year [40].

Services for MAM tended to be better prepared in agrarian areas than in pastoral regions, although there was no notable disparity in SAM readiness between the two. This could be attributed to the fact that HEP has accomplished better outcomes in agrarian regions since it was initially intended to provide healthcare packages to these areas [11,42], and has since evolved in various ways, including adaptation to pastoral communities [43]. Despite remarkable improvements in health service coverage [44], there are substantial disparities between geographic regions, urban and rural areas, and agrarian and pastoralist settings [45–48].

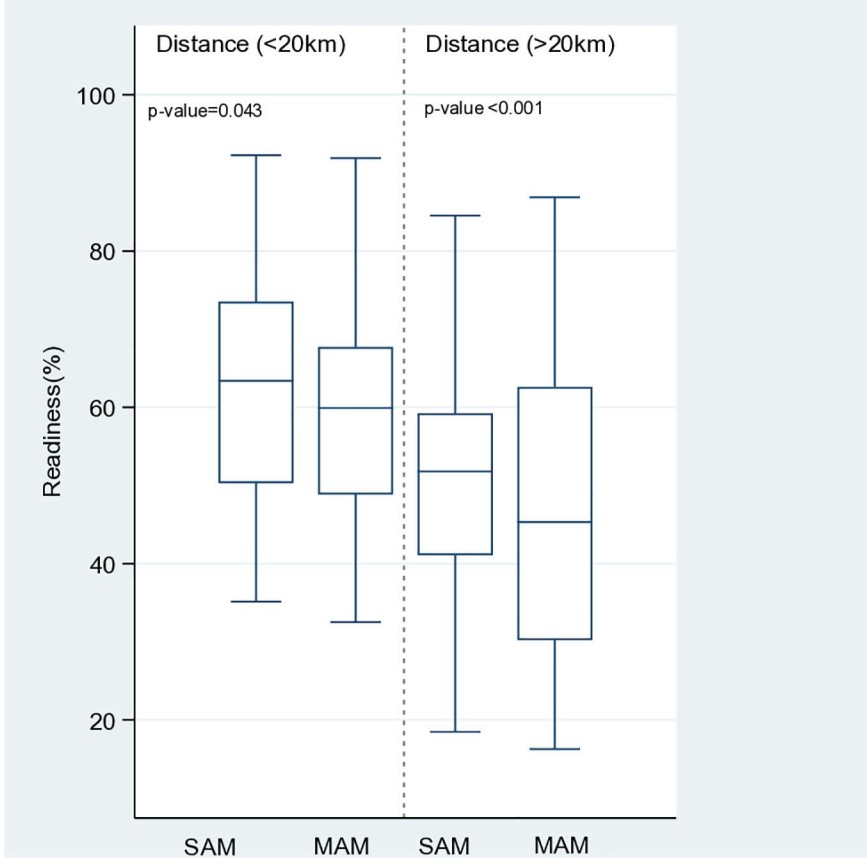

**Fig 3. Service readiness according to the distance from the catchment health center.**

Health posts located near catchment health centers were better prepared than those located further away. This could be due to the fact that nearby health posts receive more support from the nearby health center, and it is easier for the health centers to provide supervision and other necessary support to the health posts. By collaborating with each other in terms of human resources, supply, finance, and capacity building within the referral network, health facilities can enhance the quality of care, improve patient and staff satisfaction, reduce overall patient costs, and decrease preventable morbidity and mortality by reducing unnecessary referrals [49].

The study was cross-sectional; consequently, the assessment failed to capture readiness over time, which can change depending on logistics, specifically the availability of nutritional commodities at health facilities. However, this study's methodology can serve as a guide for situational analyses and policy decisions beyond the scope of this cross-sectional study. The study relied on respondents on some questions like training, with limited verification processes. As a result, recall bias may be a limitation, particularly for training questions. For instance, the survey lacked systematic registration or training reports to confirm the answers provided by health professionals. Although several health facilities in six districts across six regions were sampled, generalization was not possible particularly for health centers and hospitals. Nonetheless, we found that SAM and MAM services were unprepared, indicating lower readiness particularly at lower levels of the primary health care system.

In conclusion, the findings point to several priorities. First, strengthening the basic infrastructure at primary-level facilities is foundational and should accompany any expansion of CMAM or MAM integration into the primary health care

**Table 4. Median and third quantile regression model estimates for SAM and MAM readiness.**

| Variables | Median quantile | | Third quantile | |
|---|---|---|---|---|
| | Coefficient (CI 95%) | *p*-Value | Coefficient (CI 95%) | *p*-Value |
| Median and third quantile regression model estimates for SAM readiness | | | | |
| Type of region | | | | |
| Agrarian | 0. 0714 (0. 050, 0.193) | 0.247 | 0.107 (0.037, 0.251) | 0.143 |
| Pastoralist | 1 | | | |
| Distance from the catchment health center | | | | |
| <20km | 0.083(0.032, 0.199) | 0.155 | 0.089 (0.048, 0.226) | 0.198 |
| >20km | 1 | | | |
| Median and third quantile regression model estimates for MAM readiness | | | | |
| Type of region | | | | |
| Agrarian | 0.104(0.005, 0.205) | 0.040 | 0.160(0.026, 0.295) | 0.02 |
| Pastoralist | 1 | | 1 | |
| Distance from the catchment health center | | | | |
| <20km | 0.117(0.022, 0.211) | 0.016 | 0.042(−0.086, 0.169) | 0.517 |
| >20km | 1 | | 1 | |

system. Second, supply chain improvements and routine forecasting/prepositioning of both SAM and MAM commodities are needed for SAM and MAM to be delivered routinely at the health facilities. Third, investments to build MAM capacity through targeted training, and routine supervision are required so that MAM services are properly integrated into primary health care.

Finally, the difference in facility readiness, with some facilities performing well while many remain underprepared, underscores the need for targeted, context-specific interventions rather than one-size-fits-all solutions. Facilities in crisis-affected or remote/pastoral areas will need tailored support like mobile outreach and community-based delivery models.

## Acknowledgments

The authors wish to thank the study data collectors and district and health facility staff.

## Author contributions

**Conceptualization:** Alinoor Mohamed Farah, Seifu Hagos Gebreyesus.

**Data curation:** Alinoor Mohamed Farah.

**Formal analysis:** Alinoor Mohamed Farah.

**Funding acquisition:** Aweke Kebede, Seifu Hagos Gebreyesus.

**Methodology:** Alinoor Mohamed Farah, Seifu Hagos Gebreyesus.

**Project administration:** Alinoor Mohamed Farah, Beshada Rago, Aweke Kebede, Kemeria Barsenga, Tafara Ndumiyana, Tayech Yimer, Hiwot Darsene, Shibru Kelbessa, Beza Yilma.

**Software:** Alinoor Mohamed Farah.

**Supervision:** Samson Gebremedhin, Seifu Hagos Gebreyesus.

**Writing – original draft:** Alinoor Mohamed Farah.

**Writing – review & editing:** Samson Gebremedhin, Beshada Rago, Aweke Kebede, Seifu Hagos Gebreyesus.

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
