## [Decision Letter · Decision Letter 0]

6 Aug 2025

Dear Dr. Gebreyesus,

Thank you for submitting your manuscript to PLOS ONE. After careful consideration, we feel that it has merit but does not fully meet PLOS ONE’s publication criteria as it currently stands. Therefore, we invite you to submit a revised version of the manuscript that addresses the points raised during the review process.

We look forward to receiving your revised manuscript.

Kind regards,

Shahbaz Ahmad Zakki

Academic Editor

PLOS ONE

Journal Requirements:

“The current study is supported by World Food Program (WFP), Addis Ababa, Ethiopia”

3. In the online submission form, you indicated that “Data will be provided upon reasonable request.”

Reviewers' comments:

Reviewer's Responses to Questions

**Comments to the Author**

1. Is the manuscript technically sound, and do the data support the conclusions?

Reviewer #1: Yes

Reviewer #2: No

2. Has the statistical analysis been performed appropriately and rigorously?

Reviewer #1: Yes

Reviewer #2: No

3. Have the authors made all data underlying the findings in their manuscript fully available?

Reviewer #1: Yes

Reviewer #2: No

4. Is the manuscript presented in an intelligible fashion and written in standard English?

Reviewer #1: Yes

Reviewer #2: No

Reviewer #1: The manuscript is tehcnically sound wtihfacility-based cross-sectional design suitable for assessing the objectives. However, while statistical analysis appears appropriate, insufficient detail on sampling methods and data tools undermines methodological rigor.

Reviewer #2: Manuscript Number: PONE-D-25-08999

Article Type: Research Article

Full Title: Readiness of Health Posts to Manage Wasting in Ethiopia

Reviewer Comments-ZB

1. Title

• The title requires clarification. Consider specifying that the focus is on child wasting, if applicable.

• Additionally, explain why the study is limited to wasting rather than addressing child malnutrition more broadly as mentioned in the introduction. This will help align reader expectations with the study scope.

2. Abstract

• The abstract lacks essential components, including a clear background, study objective, methodology, and key findings.

• Please revise it to include concise information on the study rationale, design, participant population, key results, and conclusions to make it informative and self-contained.

3. Background

• The background section does not adequately contextualize child nutrition interventions at the community level. It is important to describe the expected nutrition services or packages that health post staff are mandated to implement, as per national policy.

• A brief overview of Ethiopia’s community-level healthcare system would help readers unfamiliar with the setting understand the health post’s role.

• The concept of readiness is not clearly defined or conceptualized within the context of the study. Please provide a definition and explain its dimensions.

• The study objective is vague. Clarify what aspects of readiness are being assessed—e.g., infrastructure, human resources, supplies availability, etc.

4. Methods

• The study setting should be clearly described, including geographical, administrative, and health system context.

• The description of participants is insufficient. The sentence “This survey included health posts with health extension workers (n=72) in IMAM implementing areas” does not clarify who the actual respondents were. Were health extension workers surveyed? Were any supervisors or facility heads included?

• The sample size section is unclear. Please explain how the 72 health posts were selected, what sampling method was used, and how the sample size was determined.

• The readiness assessment tool—stated to be based on the Service Availability and Readiness Assessment (SARA)—needs more elaboration. How was it adapted? What domains or indicators were included? What scoring formats were used?

• The data collection process lacks critical details: Who were the data collectors? What qualifications did they have? Was any training or supervision provided?

• Operational definitions for key terms such as wasting and readiness must be clearly provided.

• If an index score was created, specify which domains were included and how they were weighted or combined. If Principal Component Analysis (PCA) was used, provide a detailed explanation of the technique and rationale.

• The parameters in the weighted formula for index creation must be clearly defined.

• The concept of "tracer items" is referenced in the analysis but not defined in the methods. This should be explained clearly.

• The phrase “The readiness percentage score for both datasets…” is confusing. What are these datasets? Clarification is needed.

• Overall, the analysis section lacks clarity and appears to proceed without explaining the basic assumptions and analytical framework.

5. Conclusion

• Due to significant methodological and measurement gaps, particularly in defining key concepts and clearly outlining the analytical approach, the credibility of the results and conclusions is currently compromised.

• Major revisions are needed to strengthen the study’s methodological rigor and transparency to support valid conclusions.

**Do you want your identity to be public for this peer review?** For information about this choice, including consent withdrawal, please see our Privacy Policy

Reviewer #1: **Yes: ** Majid Ali Tahir

Reviewer #2: No

---

## [Author Response · Author response to Decision Letter 1]

1 Oct 2025

Reviewer 1 comments Responses

The study looks interesting and original.

Thank you for the comment

The use of a facility-based cross-sectional design is well-suited to assess service readiness.

Thank you for the comment

The author should consider making the title more specific, action-oriented and specific with area and time. Title changed/improved

The methods section of the abstract is vague. There is insufficient detail on what specific domains or indicators were used in the "weighted additive approach" to assess readiness. A brief mention of assessment tools or core domains would improve scientific transparency. Thank you for the comments. The method section of the abstract improved as per the comment.

The abstract should clarify why readiness is different for SAM vs. MAM. Without brief contextualization, the statistical difference lacks interpretive depth. Thank you for the comments on the readiness between SAM and MAM clarified.

The conclusion introduces a strong caution against rapid integration of MAM services but does not adequately reflect the findings presented. Lacking evidence of poorly planned integration in the study design. Thank you for the comment, the conclusion revised.

The authors effectively trace CMAM evolution and Ethiopia’s role, but references [1–8] are very old and back dated. Consider including more recent global reviews or evaluations post-2020 for contextual relevance. Thank for the comment, recent studies especially post 2020 included.

Sentences like “Prioritizing children at the greatest risk of mortality…” (line 96) would benefit from direct attribution to a source or clearer explanation of mortality evidence. Thank for the comment, the sentence cited and explanation of mortality evidence added.

The study objective is introduced at the very end (line 124). For clarity, move the aim earlier in the introduction and state it explicitly in one sentence. Thank for the comments, the objective introduced earlier in the introduction.

Terms like “readiness,” “wasting,” and “integration” should be defined in operational terms as used in the study context. Thank you, operational definition section included.

While the study purpose is mentioned, a formally stated research question is missing. To what extent are health posts in pilot IMAM districts prepared to manage SAM and MAM?” Thank you for the comment, the research question added.

Abbreviations are needed to be placed in appropriate manner and introduced properly (e.g., SAM, MAM), though a full glossary. Thank you for the comments, glossary of abbreviations provided.

The study states that 6 IMAM districts were chosen “purposively” (line 157) but doesn’t explain the basis for selection. Were they representative? High/low burden? More detail is needed. Thank you for the comment, detail on selection criteria added.

Within the districts, health posts were “randomly selected,” yet no method (e.g., lottery, systematic sampling) is described. Clarify the randomization process, as it contain discrepancies in sampling process Thank you for the comment. The statement revised for clarity.

The regional diversity of the districts (Afar, Amhara, Oromia, etc.) is good for generalizability (line 141), but socio-demographic profiles or burden of malnutrition in these areas would strengthen the rationale. Thank you for the comment, contextual information provided.

The tools are only called “pre-tested” (line 162) with no detail on validation or structure. It’s important to specify what domains or questions were used and if they were aligned with WHO-SARA or other standards. Thank for the comments, details on the tool structure provided.

There is no clarification about the tool, its validity and measurements. Provide detail about the tool and how it was validated and was there any approving body for that tool?

Thank you for the comments, details about the tool and the validation provided.

The variables used in regression (line 190) are mentioned vaguely. More detail is needed: e.g., was distance continuous or categorical? Was “agrarian/pastoral” self-reported? Thank your comments, all the variables were defined.

There's no mention of measures to minimize observer or selection bias (e.g., observer blinding, inter-rater reliability), or how missing data was handled. Mentioned

The results provide clear comparative statistics between SAM and MAM readiness. Percentages and tracer items are well documented. Thank you for the comment.

Figures are mentioned (Figure 1, 2), but not evaluated here. Assuming clarity, they complement the tabular results well. Detailed narration provided for each figure

Statements like “p < 0.001” (line 230) are given, but interpretation is minimal. Authors should explain implications—e.g., what does a statistically significant readiness gap mean in service delivery? Interpretation provided.

Results are presented as data points mostly already in table and figure form. There is need to provide the contextual information in the results section as well. For example, staff training is high (88.9%) but equipment availability is low (e.g., timers 16.7%). Such typer of arguments and contrasts are not highlighted. Thank you for comments, the result for this section is detailed as recommended.

The disaggregation by pastoral and agrarian areas is highly relevant, though details in results are thin. Include summary statistics (e.g., exact readiness percentages in pastoral areas). Summary statistics included.

The availability of commodities (67–69%) contrasts with basic amenities like electricity (40.3%). This should be flagged more clearly. Flagged

The table shows weighted scores but lacks 95% confidence intervals for domain-level results, which would improve interpretation of variability. CI provided

The discussion lacks comparative benchmarking. How do Ethiopia's results compare to readiness levels in similar LMIC contexts? This limits the paper’s global relevance. Discussion section improved and the study results discussed in relation to similar contexts.

The discussion and analysis remains descriptive, not analytical. There is no in-depth discussion of why infrastructure gaps persist or which system bottlenecks are most critical. Use of a health systems framework (e.g., WHO building blocks) would have added rigor. The discussion improved

The authors acknowledge cross-sectional limitations but should also critique data reliability more strongly. The reliance on self-reported data with “limited verification” (line 354) is a major methodological weakness and reduces the study’s credibility. The limited verification was for only one question on training as there were no means to verify.

The conclusion restates key findings on low readiness effectively. However, provide little or no offer specific policy or programmatic recommendations based on findings. For example, what type of integration approach or capacity-building measures would be feasible given current resource constraints? Conclusion improved.

Reviewer 2 comments Responses

The title requires clarification. Consider specifying that the focus is on child wasting, if applicable. Title improved based on the comments.

Additionally, explain why the study is limited to wasting rather than addressing child malnutrition more broadly as mentioned in the introduction. This will help align reader expectations with the study scope. Explained under the introduction section

The abstract lacks essential components, including a clear background, study objective, methodology, and key findings. Abstract improved and the mission component added.

Please revise it to include concise information on the study rationale, design, participant population, key results, and conclusions to make it informative and self-contained. Revised

The background section does not adequately contextualize child nutrition interventions at the community level. It is important to describe the expected nutrition services or packages that health post staff are mandated to implement, as per national policy. Mentioned briefly

A brief overview of Ethiopia’s community-level healthcare system would help readers unfamiliar with the setting understand the health post’s role Mentioned under study setting

The concept of readiness is not clearly defined or conceptualized within the context of the study. Please provide a definition and explain its dimensions. Explained under method section

The study objective is vague. Clarify what aspects of readiness are being assessed—e.g., infrastructure, human resources, supplies availability, etc. Detailed explained under method section

The study setting should be clearly described, including geographical, administrative, and health system context. The study setting revised and well described

The description of participants is insufficient. The sentence “This survey included health posts with health extension workers (n=72) in IMAM implementing areas” does not clarify who the actual respondents were. Were health extension workers surveyed? Were any supervisors or facility heads included? Participants were described.

The sample size section is unclear. Please explain how the 72 health posts were selected, what sampling method was used, and how the sample size was determined. Revised and explained

The readiness assessment tool—stated to be based on the Service Availability and Readiness Assessment (SARA)—needs more elaboration. How was it adapted? What domains or indicators were included? What scoring formats were used? Revised and described

The data collection process lacks critical details: Who were the data collectors? What qualifications did they have? Was any training or supervision provided? The collection process improved and more details incorporated.

Operational definitions for key terms such as wasting and readiness must be clearly provided. Operation definition provided

If an index score was created, specify which domains were included and how they were weighted or combined. If Principal Component Analysis (PCA) was used, provide a detailed explanation of the technique and rationale. Explained and table provided

The parameters in the weighted formula for index creation must be clearly defined. Defined

The concept of "tracer items" is referenced in the analysis but not defined in the methods. This should be explained clearly.

Defined and included a table

The phrase “The readiness percentage score for both datasets…” is confusing. What are these datasets? Clarification is needed. Revised, it was a typo error

Overall, the analysis section lacks clarity and appears to proceed without explaining the basic assumptions and analytical framework. This section improved with explanations provided

---

## [Editor Report · Decision Letter 1]

5 Oct 2025

Readiness of Health Posts to Manage Child Wasting in Six Districts of Ethiopia.

PONE-D-25-08999R1

Dear Dr. Seifu Hagos Gebreyesus,

We’re pleased to inform you that your manuscript has been judged scientifically suitable for publication and will be formally accepted for publication once it meets all outstanding technical requirements.

Kind regards,

Shahbaz Ahmad Zakki

Academic Editor

PLOS ONE
---

## [Editor Report · Acceptance letter]

PONE-D-25-08999R1

PLOS ONE

Dear Dr. Gebreyesus,

I'm pleased to inform you that your manuscript has been deemed suitable for publication in PLOS ONE. Congratulations! Your manuscript is now being handed over to our production team.

Kind regards,

on behalf of

Dr. Shahbaz Ahmad Zakki

Academic Editor

PLOS ONE